# Effects of Micro-Arrangement of Solid Particles on PCE Migration and Its Remediation in Porous Media

Ming Wu[1,2], Jianfeng Wu [1*], Jichun Wu[1**], and Bill X. Hu[2]

[1]Key Laboratory of Surficial Geochemistry, Ministry of Education; Department of

Hydrosciences, School of Earth Sciences and Engineering, Nanjing University, Nanjing

210023, China

[2]Institute of Groundwater and Earth Sciences, Jinan University, Guangzhou 510632,

China

*Correspondence to*: J.F. Wu (jfwu@nju.edu.cn), J. C. Wu (jcwu@nju.edu.cn)

**ABSTRACT**

Groundwater can be stored abundantly in granula-composed aquifers with high permeability. The micro-structure of granular materials has important effect on the permeability of aquifers; and the contaminant migration and remediation in aquifers is also influenced by the characteristics of porous media. In this study, two different microscale arrangements of sand particles are compared to reveal the effects of micro-structure on the contaminant migration and remediation. With the help of fractal theory, the mathematical expressions of permeability and entry pressure are conducted to delineate granular materials with regular triangle arrangement (RTA) and square pitch arrangement (SPA) at microscale. Using Sequential Gaussian Simulation (SGS) method, a synthetic heterogeneous site contaminated by Perchloroethylene (PCE) is then used to investigate the migration and remediation affected by the two different micro-scale arrangements. PCE is released from an underground storage tank into the aquifer and the surfactant is used to clean up the subsurface contamination. Results suggest that RTA can not only cause larger range of groundwater contamination, but also make remediation become more difficult. The PCE remediation efficiency of 60.01% -99.78% with a mean of 92.52% and 65.53% -99.74% with a mean of 95.83% are achieved for 200 individual heterogeneous realizations based on the RTA and SPA, respectively, indicating that the cleanup of PCE in aquifer with SPA is significantly easier. This study leads to a new understanding of the microstructures of porous media and demonstrates how micro-scale arrangements control contaminant migration in aquifers, which is helpful to design

successful remediation scheme for underground storage tank spill.
**Keywords:** microscale arrangement; regular triangle; square pitch; contaminant
migration and remediation; cumulative PCE removal; macroscopic scale

# 1. Introduction

Groundwater is an essential natural resource for water supply to domestic,

agricultural, industrial activities and ecosystem health (Boswinkel, 2000; Valipour, 2012;
Valipour, 2015; Yannopoulos et al., 2015; Valipour and Singh, 2016). Unfortunately, with
the rapid development of economic activities such as mining, agriculture, landfills and
industrial activities (Bakshevskaia and Pozdniakov, 2016; Cui et al., 2016; Liu et al.,
2016; An et al., 2017; Shen et al., 2017), more and more contaminants released from
human activities are contaminating the precious groundwater resource and subsurface
environment (Dawson and Roberts, 1997; Liu, 2005; Hadley and Newell, 2014;
C.Carroll et al., 2015; Essaid et al., 2015; Huang et al., 2015; Liu et al., 2016; Schaefer et
al., 2016; Weathers et al., 2016). Among the contaminants detected in groundwater,
dense nonaqueous phase liquids (DNAPLs) such as perchloroethylene (PCE) and other
polycyclic aromatic hydrocarbons (PAHs), are highly toxic and carcinogenic (Dawson
and Roberts, 1997; Hadley and Newell, 2014). When DNAPLs are released into aquifer
from underground storage tank, they will infiltrate through the entire aquifer and form
residual ganglia and pools of DNAPLs due to their large densities, high interfacial
tension, and low solubility. The residual ganglia and pools of DNAPLs can serve as
long-term sources of groundwater contamination which are harmful to subsurface
environment and human beings (Bob et al., 2008; Liang and Lai, 2008; Liang and Hsieh,
2015). Consequently, it is very important to explore DNAPL migration in aquifer and
associated remediation of groundwater contamination.
When DNAPLs migrate in aquifers at macroscopic scale, the transport properties
such as permeability, diffusivity and dispersivity are closely related to the aquifer's
microstructures and can affect DNAPLs behavior (Yu and Li, 2004; Yu, 2005; Yun et
al., 2005; Feng and Yu, 2007; Yu et al., 2009). Therefore, characterizing the effect of
microstructures on macroscopic properties is a key point of research on heterogeneity
of porous media (Mishra et al., 2016). In the classical Kozeny–Carman equation, the
permeability $K$ is related to porosity $n$, surface area $S$ and the Kozeny constant $c$,
where $c$ is affected by the porosity, solid particles and micro geometric structures
(Bear 1972; Yu et al. 2009). According to fractal theory, natural porous media can be
treated as fractal objects (Pfeifer and Avnir 1983; Katz and Thompson 1985; Krohn
1988). For example, the tortuosity of flow path in porous media is deeply studied by
various proposed fractal models (Yu and Cheng 2002; Yu et al. 2009; Cai et al. 2010),
indicating the effectiveness of fractal methods. Based on fractal concepts, mathematic
models are proposed to depict the permeability and invasion of fluids in some special
porous media (Yu and Cheng 2002; Yu et al. 2009; Cai et al. 2010). Furthermore, fractal
method is also used to explore the effect of microstructure of biological media on
associated thermal conductivity while this kind of material has a complex randomly
distributed vascular trees structure at microscale (Li and Yu 2013).
In this study, we focus on the effect of micro-arrangement of sand particles on
macroscopic DNAPL migration and associated remediation for underground storage
tank spill. With the help of fractal theory, the microstructures of two different microscale
arrangements of sand particles are explored. Afterwards, the mathematical relationships
between porosity and permeability, entry pressure are derived for regular triangle
arrangement (RTA) and square pitch microscale arrangement (SPA). Idealized
heterogeneous contaminated site is generated using Sequential Gaussian Simulation
(SGS) method. Underground storage tank releases PCE into heterogeneous aquifer
composed of granular material. After long time migration, PCE contamination is
alleviated using surfactant remediation method. A multicomponent, multiphase model
simulator UTCHEM is then used to simulate the entire process of DNAPL migration and
remediation. Effects of arrangements of sand particles on migration and remediation of
DNAPLs are comparatively analyzed based on the simulations to reveal how the
microstructure of porous media controls the contaminant migration and remediation at
macroscopic scale.
**2. Methodology**
**2.1 Fractal models of two different microscale arrangements of sand**
**particles**
The porous media can be treated as the bundle of tortuous capillary tubes, the
relationship between the diameter and the length of capillary tube are (Yu and Cheng,

2002):

$$L_t(\lambda) = \lambda^{1-D_t} L_s^{D_t} \tag{1}$$
where $L_s$ is the straight length between the tortuous flow path's end point; $\lambda$ is the
diameter of capillary tube; $D_t$ is the fractal dimension of tortuosity for porous media,
$1<D_t<2$ (Yu and Cheng, 2002).
Select an infinitesimal element consisting of a bundle of tortuous capillary tubes
form porous media, the total number of capillary tubes in infinitesimal element can be
calculated by the power-law relation:

$$N(L \geq \lambda) = (\frac{\lambda_{\max}}{\lambda})^{D_f} \tag{2}$$


where $D_f$ is the fractal dimension for pore areas in porous media, $1<D_f<2$ (Yu and
Cheng, 2002); $\lambda_{max}$ is the maximum diameter of capillary tubes.
Afterward, the derivative of Equation (2) can be achieved:

$$-dN(L \geq \lambda) = D_f \lambda_{\max}^{D_f} \lambda^{-(D_f+1)} d_\lambda \tag{3}$$


The total number of capillary tubes in infinitesimal element can be derived from
Equation (3):

$$N_t(L \geq \lambda_{\min}) = (\frac{\lambda_{\max}}{\lambda_{\min}})^{D_f} \tag{4}$$


where $\lambda_{min}$ is the minimum diameter of capillary tubes.
Dividing Equation (3) by Equation (4) can achieve:

$$-\frac{d_{N(L \geq \lambda)}}{N_t} = D_f \lambda_{\min}^{D_f} \lambda^{-(D_f+1)} d_\lambda = f(\lambda) d_\lambda \tag{5}$$


where $f(\lambda)$ is the probability density function, $f(\lambda) = D_f \lambda_{\min}^{D_f} \lambda^{-(D_f+1)}$.
The probability density function satisfies the relationship:

$$\int_{-\infty}^{+\infty} f(\lambda) d_\lambda = 1 - (\frac{\lambda_{\min}}{\lambda_{\max}})^{D_f} \tag{6}$$


Considering $(\frac{\lambda_{min}}{\lambda_{max}})^{D_f} = 0$, the above Equation (6) becomes:
$$\int_{-\infty}^{+\infty} f(\lambda)d_\lambda = \int_{\lambda_{min}}^{\lambda_{max}} f(\lambda)d_\lambda = 1 - (\frac{\lambda_{min}}{\lambda_{max}})^{D_f} = 1 \qquad (7)$$
When fluid flow in capillary tubes, the flow rate $Q$ can be calculated by the
Hagen–Poiseulle equation:
$$Q = \frac{\pi r^4 \Delta P}{8\mu L_s} = \frac{\pi (\frac{\lambda}{2})^4 \Delta P}{8\mu L_s} = \frac{\pi \lambda^4 \Delta P}{128\mu L_s} \qquad (8)$$
where $\mu$ is fluid's viscosity; $\Delta P$ is the pressure gradient across the capillary tube.
The differentiation of flow rate of capillary tubes is (Yu and Cheng, 2002):
$$d_q = [-d_{N(L\geq\lambda)}]\frac{\pi\lambda^4\Delta P}{128\mu L_t(\lambda)} = D_f\lambda_{max}^{D_f}\lambda^{-(D_f+1)}d_\lambda \cdot \frac{\pi\lambda^4\Delta P}{128\mu L_t(\lambda)}$$
$$= \frac{\pi}{128}\frac{\Delta P}{\mu}\frac{D_f\lambda_{max}^{D_f}}{L_t(\lambda)}\lambda^{3-D_f}d_\lambda = \frac{\pi}{128}\frac{\Delta P}{\mu}\frac{D_f\lambda_{max}^{D_f}}{\lambda^{1-D_t}L_s^{D_t}}\lambda^{3-D_f}d_\lambda \qquad (9)$$
$$= \frac{\pi}{128}\frac{\Delta P}{\mu}\frac{D_f\lambda_{max}^{D_f}}{L_s^{D_t}}\lambda^{2+D_t-D_f}d_\lambda$$

Integrating the individual flow rate from $\lambda_{min}$ to $\lambda_{max}$ can achieve the total flow rate
(Yu and Cheng, 2002):
$$Q = \int d_q = \int_{\lambda_{min}}^{\lambda_{max}}\frac{\pi}{128}\frac{\Delta P}{\mu}\frac{D_f\lambda_{max}^{D_f}}{L_s^{D_t}}\lambda^{2+D_t-D_f}d_\lambda$$
$$= \frac{\pi}{128}\frac{\Delta P}{\mu}\frac{D_f}{3-D_f+D_t}\frac{1}{L_s^{D_t}}\lambda_{max}^{D_f}(\lambda_{max}^{3-D_f+D_t} - \lambda_{min}^{3-D_f+D_t}) \qquad (10)$$
$$= \frac{\pi}{128}\frac{\Delta P}{\mu}\frac{D_f}{3-D_f+D_T}\frac{1}{L_s^{D_t}}\lambda_{max}^{3+D_t}[1 - (\frac{\lambda_{min}}{\lambda_{max}})^{D_f}(\frac{\lambda_{min}}{\lambda_{max}})^{3+D_t-2D_f}]$$

Due to $1<D_t<2$ and $1<D_f<2$, then $3+D_T-2D_f>0$. Simultaneously, $(\frac{\lambda_{min}}{\lambda_{max}})^{D_f} \cong 0$,
$0 < (\frac{\lambda_{min}}{\lambda_{max}})^{3+D_T-D_f} < 1$. Therefore, Equation (10) can be simplified as:
$$Q = \int d_q = \frac{\pi}{128} \frac{\Delta P}{\mu} \frac{D_f}{3 - D_f + D_T} \frac{1}{L_0^{D_T}} \lambda_{\max}^{3+D_T} \qquad (11)$$

Substituting Darcy's law $Q = \dfrac{kA\Delta P}{\mu L_0}$ in Equation (11) will obtain the permeability
of porous media:
$$k = \frac{\pi}{128} \frac{D_f}{3 + D_T - D_f} \frac{L_0^{1-D_T}}{A} \lambda_{\max}^{3+D_T} \qquad (12)$$

To obtain the fractal dimension of tortuosity $D_t$, the expression of tortuosity $(\tau)$
can be obtained from Equation (1):
$$\tau = \frac{L_t(\lambda)}{L_s} = \frac{\lambda^{1-D_t} L_s^{D_t}}{L_s} = (\frac{L_s}{\lambda})^{D_t - 1} \qquad (13)$$

Then $D_t$ is given by (Yu and Li, 2001):
$$D_{\mathrm{t}} = 1 + \frac{\ln \tau}{\ln(\dfrac{L_s}{\lambda})} \qquad (14)$$

RTA and SPA are shown in Fig. 1. An equilateral triangle and a square are
selected from the two micro-structures as unit cells (Fig. 1a and Fig. 1b). The unit cell
of equilateral triangle is composed of three solid particles and the pore among them,
while the unit cell of square is composed of four solid particles. For the unit cell of
RTA in Fig. 1a, corresponding porosity is given by:
$$n = \frac{A_a - \pi R_v^2 / 2}{A_a} \qquad (15)$$

where $n$ is porosity; $A_a$ is the total area of equilateral triangle; $R_v$ is the average radius
of solid particles. The total area of equilateral triangle can be achieved:
$$A_a = \frac{\pi R_v^2}{2(1 - n)} \qquad (16)$$

The side length of the equilateral triangle in Fig. 1a can be calculated as:
$$L_a = R_v \sqrt{\frac{2\pi}{\sqrt{3}(1-n)}} \qquad (17)$$

where $L_a$ is the side length.
The area of irregular pore among solid particles is given by:
$$A_{ap} = A_a - \frac{\pi R_v^2}{2} = \frac{\pi R_v^2 n}{2(1-n)} \qquad (18)$$

where $A_{ap}$ is the area of pore in the unit cell.
Approximate the pore in the equilateral triangle as a circle, then the maximum
diameter of pore can be obtained:
$$\lambda_{max,a} = R_v \sqrt{\frac{2n}{1-n}} \qquad (19)$$

where $\lambda_{max,a}$ is the diameter of capillary tube in equilateral triangle. The fluid does not
only pass the central-pore of the unit cell, but also flow through the gap between
adjacent particles. The gap length and the average diameter of capillary tube
perpendicular to the plane of equilateral triangle are calculated as follows:
$$\Delta L_a = L_a - 2R_v = R_v \left( \sqrt{\frac{2\pi}{\sqrt{3}(1-n)}} - 2 \right) \qquad (20)$$

$$\lambda_a = \frac{\lambda_{max,a} + \Delta L_a}{2} = \frac{R_v}{2} \left( \sqrt{\frac{2n}{1-n}} + \sqrt{\frac{2\pi}{\sqrt{3}(1-n)}} - 2 \right) \qquad (21)$$

where $\Delta L_a$ is the gap length between solid particles; $\lambda_a$ is the average diameter of
capillary tubes in the equilateral triangle.
Generally, the tortuosity of flow path in porous media is the ratio of the length of
tortuous flow path to the straight length of flow path along the flow direction (Taiwo et al.,

2016):

$$\tau = \frac{L_t}{L_s} \qquad (22)$$

where $L_t$ is the length of tortuous flow path; and $L_s$ is the straight length of flow path
along the flow direction.
For the flow path shown in Fig. 1a, the $L_t$ and $L_s$ respectively are:

$$L_t = (h_o - R_v) + \frac{\pi R_v}{2} = R_v \left( \sqrt{\frac{\sqrt{3}\pi}{2(1-n)}} + \frac{\pi}{2} - 1 \right) \tag{23}$$

$$L_s = h_o = R_v \sqrt{\frac{\sqrt{3}\pi}{2(1-n)}} \tag{24}$$

where $h_o$ is the altitude of the equilateral triangle, $h_o = \frac{\sqrt{3}}{2} L_a = R_v \sqrt{\frac{\sqrt{3}\pi}{2(1-n)}}$ .
Consequently, the tortuosity of RTA is yielded:

$$\tau = \frac{L_t}{L_s} = 1 + \frac{\frac{\pi}{2} - 1}{\sqrt{\frac{\sqrt{3}\pi}{2(1-n)}}} \tag{25}$$

The $D_f$ is determined using Sierpinkski gasket (Fig. 2) in fractal theory (Yu and
Cheng, 2002). The shaded area represents solid of porous media and the white area
represents pore. The pore area fractal dimension in Figs. 2a-c are 0.000, 1.000 and 1.594,
respectively ($1 = L_a^{D_f} = 2^{D_f}$, $3 = L_a^{D_f} = 3^{D_f}$, $13 = L_a^{D_f} = 5^{D_f}$). Based on the Sierpinkski gasket,
the dimensionless pore area in RTA (Fig. 1a) is approximated as:

$$A_{apd} = (L_a^+)^{D_f} \tag{26}$$

where $A_{apd}$ is the dimensionless pore area of RTA; $L_a^+ = L_a / \lambda_{min}$ . Equation (26) can be
solved to achieve $D_f$:

$$D_f = \frac{\ln A_{apd}}{\ln L_a^+} \tag{27}$$

The porosity equals to the ratio of the dimensionless pore area of RTA ($A_{apd}$) to
the dimensionless total area of RTA ($A_a^+$):
$$n = \frac{A_{apd}}{A_a^+}$$
(28)

where $A_a^+ = \dfrac{A_a}{\pi\lambda_{\min}^2/4} = \dfrac{\dfrac{\pi R_v^2}{2(1-n)}}{\pi\dfrac{\lambda_{\min}^2}{4}} = \dfrac{2R_v^2}{\lambda_{\min}^2}\dfrac{1}{1-n} = \dfrac{(d^+)^2}{2}\dfrac{1}{1-n}\,; d^+ = \dfrac{2R_v}{\lambda_{\min}}\,,\quad L_a^+ = \sqrt{A_a^+}\,.$

From Equation (28), the dimensionless pore area of RTA ($A_{apd}$) is given by:

$$A_{apd} = n \cdot A_a^+$$
(29)

The dimensionless total area of RTA ($A_a^+$) can be written as:

$$A_a^+ = (L_a^+)^2$$
(30)

Afterward, $L_a^+$ is calculated as:

$$L_a^+ = \sqrt{A_a^+} = \sqrt{\frac{(d^+)^2}{2}\frac{1}{1-n}} = d^+\sqrt{\frac{1}{2(1-n)}}$$
(31)

Substituting Equation (29) and Equation (31) into Equation (27) will derive $D_f$ of

RTA:

$$D_f = \frac{\ln A_{apd}}{\ln L_a^+} = \frac{\ln(n\cdot A_a^+)}{\ln(\sqrt{A_a^+})} = 2 + \frac{\ln(n)}{\ln(\sqrt{A_a^+})} = 2 + \frac{\ln(n)}{\ln(d^+\sqrt{\dfrac{1}{2(1-n)}})}$$
(32)

For the unit cell of square shown in Fig. 1b, the porosity is:

$$n = \frac{A_b - \pi R_v^2}{A_b}$$
(33)

where $A_b$ is the total area of the square. Equation (33) can also be expressed as the

area of unit cell:

$$A_b = \frac{\pi R_v^2}{1-n}$$
(34)

Again, the side length of the square is:

$$L_b = \sqrt{A_b} = R_v\sqrt{\frac{\pi}{1-n}}$$
(35)

Consequently, the area of irregular pore in the square is given by:

$$A_{bp} = A_b - \pi R_v^2 = \frac{n\pi R_v^2}{1-n} \tag{36}$$

where $A_{bp}$ is the area of pore in the square.
Approximate the pore as a circle and obtain corresponding maximum diameter:

$$\lambda_{max,b} = 2R_v\sqrt{\frac{n}{1-n}} \tag{37}$$

where $\lambda_{max,b}$ is the maximum diameter of capillary tube perpendicular to the plane of
the square. Similarly, fluid flows through the central-pore in the square and the gap
between adjacent particles. As a result, the gap and average diameter of capillary tube
are expressed as:

$$\Delta L_b = L_b - 2R_v = R_v\left(\sqrt{\frac{\pi}{1-n}} - 2\right) \tag{38}$$

$$\lambda_b = \frac{\lambda_{max,b} + \Delta L_b}{2} = \frac{R_v}{2}\left(2\sqrt{\frac{n}{1-n}} + \sqrt{\frac{\pi}{1-n}} - 2\right) \tag{39}$$

where $\Delta L_b$ is the gap length between the adjacent two solid particles; $\lambda_b$ is the average
diameter of capillary tube.
For the tortuous flow path in Fig. 1b, the $L_t$ and $L_s$ respectively are given by:

$$L_t = \Delta L_b + \pi R_v = R_v\left(\sqrt{\frac{\pi}{1-n}} - 2 + \pi\right) \tag{40}$$

$$L_s = L_b = R_v\sqrt{\frac{\pi}{1-n}} \tag{41}$$

Afterward, the tortuosity of SPA yields:

$$\tau = \frac{L_t}{L_s} = 1 + \frac{\pi-2}{\sqrt{\dfrac{\pi}{1-n}}} \tag{42}$$

The procedure of deriving $D_f$ of SPA is similar to the procedure of calculating $D_f$ of
RTA. Similarly, the $D_f$ and porosity of SPA (Fig. 1b) are given by:
$$D_f = \frac{\ln A_{bpd}}{\ln L_b^+}$$
(43)

$$n = \frac{A_{bpd}}{A_b^+}$$
(44)

where $A_{bpd}$ is the dimensionless pore area of SPA; $L_b^+ = L_b / \lambda_{min}$, $A_b^+$ is the
dimensionless total area of SPA, $A_b^+ = \dfrac{A_b}{\pi \lambda_{min}^2 / 4} = \dfrac{\dfrac{\pi R_v^2}{1-n}}{\pi \dfrac{\lambda_{min}^2}{4}} = \dfrac{4 R_v^2}{\lambda_{min}^2} \dfrac{1}{1-n} = (d^+)^2 \dfrac{1}{1-n}$.
The dimensionless pore area of SPA ($A_{bpd}$) can be yielded from Equation (44):
$$A_{bpd} = n \cdot A_b^+$$
(45)

$L_b^+$ can be calculated as:
$$L_b^+ = \sqrt{A_b^+} = \sqrt{(d^+)^2 \frac{1}{1-n}} = d^+ \sqrt{\frac{1}{1-n}}$$
(46)

Substituting Equation (45) and Equation (46) into Equation (43), $D_f$ of SPA can be
derived:
$$D_f = \frac{\ln A_{bpd}}{\ln L_b^+} = \frac{\ln(n \cdot A_b^+)}{\ln(\sqrt{A_b^+})} = 2 + \frac{\ln(n)}{\ln(\sqrt{A_b^+})} = 2 + \frac{\ln(n)}{\ln(d^+ \sqrt{\dfrac{1}{1-n}})}$$
(47)

The entry pressure of tortuous capillary tube ($P_c$) is defined by Young-Laplace
equation as follows ( Ahn and Seferis, 1991):
$$P_c = \frac{\omega}{\lambda} \frac{1-n}{n}$$
(48)

where $P_c$ is the entry pressure; $\lambda$ is the diameter of capillary tube; $\omega$ equals to $F\sigma\cos\theta$
in which $\theta$ is the contact angle between fluid and solid, $\sigma$ is the surface tension of the
wetting fluid, and $F$ is the form factor depending on the capillary tube alignment and
the flow direction.

## 2.2 Dealing with the heterogeneity of porous media


In this study, Sequential Gaussian Simulation (SGS) is used to generate random


realization of heterogeneous porosity field. SGS is a stochastic simulation method


combining sequential principle and Gaussian method. It assumes variable fit to Gaussian


random field. The gauss distribution function is constructed at the each simulated spatial


location based on the characteristics of variation function, afterward, randomly selects a


value as the variable at the location. In SGS method, observation data are transformed to


Gaussian distribution or normal distribution. Based on current sample data, the


conditional probability distribution of points to be simulated is calculated by SGS


method and then simulation is performed based on semivariogram model. Each


simulated value, together with measured data and previous simulation data, becomes the


conditional data set for the next step. As simulation proceeds, the conditional data set


increases. Pervious researches suggest 50–400 realizations are required to obtain a


statistically stable mean realization (Eggleston et al., 1996; Hu et al., 2007).


## 2.3 Modeling PCE migration and its remediation


The DNAPL migration and remediation are modeled using a multi-component,


multi-phase, and multi-composition simulator named UTCHEM (University of Texas


Chemical Compositional Simulator) (Delshad et al., 1996). As an extension to Delshad's


work, UTCHEM was developed by University of Texas as a comprehensive and practical


tool. In numerous applications, UTCHEM has proved to be particularly useful in


simulation of contaminant migrations and has been a popular multi-phase flow,


multi-constituent, reactive transport model used widely in groundwater simulations.
UTCHEM account for chemical, physical and biological reactions, complex
non-equilibrium sorption, decay and geochemical reactions, surfactant-enhanced
solubilization and mobilization of DNAPLs. Moreover, heterogeneous properties of
porous media is also considered. As a result, UTCHEM has been adapted for a variety of
environmental applications such as surfactant-enhanced aquifer remediation (SEAR). In
this study, DNAPL migration and remediation for cleaning up DNAPL contamination in
idealized heterogeneous site are simulated by UTCHEM.

## 3. Application to a synthetic heterogeneous PCE contaminated site

### 3.1 Site description

The idealized domain synthetic application is a two-dimensional confined aquifer
(Fig. 3). The length, width and depth of aquifer are 101 m, 25 m and 25 m, respectively.
Idealized aquifer is discretized into 101 grids horizontally and 25 layers vertically (Fig.
3b). The spacing of each grid is uniformly 1 m along $x$ and $z$ directions, and the
longitudinal and transverse dispersivities are set as to 1.0 m and 0.1 m, respectively.
Horizontal and vertical correlation length values is 5 m. The top and bottom borders of
aquifer are defined as no-flow boundaries, while the left and right borders are defined as
constant potential boundaries to create a groundwater flow from left to right under a low
hydraulic gradient of 0.005 m/m (Liu et al., 2003; Liu, 2005; Qin et al., 2007). The
porous media of idealized aquifer is assumed to be heterogeneous and mixed by different
grades of sands.
The porosity of aquifer is assumed spatially and uniformly distributed with average
value of 0.220 and standard deviation of 0.060. In this study, porosity follows normal
distribution and its standard deviation (SD) represents the enhanced geological
heterogeneity. 200 realizations porosity field are generated using Sequential Gaussian
Simulation (SGS). One of the 200 realizations of heterogeneous field is shown in Fig. 4a.
Simultaneously, statistical assessment is taken on the individual realization of porosity
field and corresponding histograms are shown in Fig. 4b. We can find the frequency of
the individual realization of porosity field is close to normal distribution, which conform
to the situation that most characteristic of natural aquifer can be expressed as normal
distribution (Montgomery et al, 1987). Based on the heterogeneous porosity field, the
fractal dimension of tortuosity $D_t$, the fractal dimension for pore areas $D_f$ and the
diameter of capillary tube in porous media, permeability is obtained by the Equation
(12). Fig. 4c shows the individual heterogeneous permeability field selected from the
200 realizations of RTA, besides, the result of associated frequency analysis is shown in
Fig. 4d. The permeability field fits the lognormal distribution obviously, which has been
presented by many researches that the parameter of aquifer penetrability follows
lognormal distribution (Montgomery et al., 1987; Veneziano and Tabaei, 2004).
Compared to histogram of porosity field in Fig. 4b, the shape of permeability is similar.
The individual heterogeneous permeability field of SPA is shown in Fig. 4e.
Corresponding frequency analysis of SPA reveals the permeability field is lognormal
distribution, while some difference appears compared with RTA (Fig. 4f). The average
permeability of individual realization of RTA is $2.012 \times 10^{-12}$ m$^2$ and the average
permeability of individual realization of SPA is $1.618 \times 10^{-12}$ m$^2$. For 200 realizations, the
average permeability of RTA and SPA are $2.120 \times 10^{-12}$ m$^2$ and $1.706 \times 10^{-12}$ m$^2$, indicating
the permeability of RTA is bigger than SPA slightly.

The average pore diameters of two different microscale arrangements of particles

are derived using corresponding fractal models. In detail, average diameter of RTA is
calculated by Equation (21) and average diameter of SPA is calculated by Equation
(39). Consequently, the entry pressure of the two kinds of microscale arrangements
can be obtained by Equation (48), respectively. The individual entry pressure fields of
two microscale arrangements and associated frequency analysis are shown in Figs. 4g-j.
From the frequency of entry pressure in Fig. 4h and Fig. 4j, the entry pressures of both
RTA and SPA are the lognormal distributions. However, the average entry pressure of
individual realization of RTA is 1.980 kPa, while the average entry pressure of SPA is
1.481 kPa. For 200 realizations of entry pressure field, the average entry pressure of RTA
is 1.922 kPa and the average entry pressure of SPA is 1.442 kPa. The differences of
average entry pressure between RTA and SPA imply the micro-structure of aquifer has
effect on the macroscopic characteristics.

The purpose of this study is to explore the effects of micro-structure of aquifer on

DNAPL migration and remediation. A PCE spill event (the leaking of underground
storage tank) occurs on the top of the aquifer and a surfactant remediation is designed to
clean up the contaminated aquifer. The total duration of 300 days is divided into four
stages: (1) 300 m$^3$ PCE is released from underground storage tank into aquifer at the top
layer of spill position shown in Fig. 3a during 0~30 days; (2) PCE migrates in aquifer
freely during 30~100 days; (3) surfactant is injected into aquifer during 100~150 days;
and (4) water flushing during 150~300 days. In the first stage, PCE is released as a point
pollution source in the center grid block at the top layer of the aquifer, which spill is at a
constant rate of 10 $m^3$/day. After PCE coming into heterogeneous aquifer, PCE is
migrating freely under the effects of gravity and the natural hydraulic gradient condition.
The PCE not only migrates downward through the aquifer, but also can be trapped by
capillary forces as residual ganglia and globules. During the long-term PCE migration
period, PCE is contaminating groundwater and expanding plume. To clean up the
contaminated aquifer, 4% surfactant solution is injected into aquifer through the two
injection wells (Fig. 3b) at a constant rate of 80 $m^3$/day, simultaneously, contaminated
groundwater is extracted through production well at constant rate of 160 $m^3$/day.
Surfactant can reduce the interfacial tension between DNAPL and aqueous phase to
promote solubilization and mobilization of DNAPL. After surfactant injection, the
contaminated aquifer is flushed by water over a long time of 150 days. Based on the
distributions of porosity, permeability and entry pressure of two microscale arrangements,
the entire PCE migration and remediation process is simulated by a multicomponent,
multiphase model simulator UTCHEM (Delshad et al., 1996). The parameters used in
simulation are listed in Table 1. Simulation results of two different microscale
arrangements are compared to reveal the effect of microstructure on the DNAPL
migration and remediation.

## 3.2 Results and discussion

### 3.2.1 PCE migration and its remediation based on single realizations

The simulation results of PCE migration for individual realization of porosity field for RTA are shown in Fig. 5a-f. When PCE is released into aquifer into aquifer at the top layer of spill position, PCE almost infiltrates vertically under the effect of gravity force (Fig. 5a). Due to the heterogeneity of aquifer, some preferential flow appears and PCE plume becomes irregular (Fig. 5b). After 30 days, PCE plume almost touches the bottom of aquifer (Fig. 5c). When the PCE leakage is stopped, PCE migrates continuously in aquifer for 70 days (Fig. 5d-f). The released PCE is migrating downward and entrapped by capillary forces as residual ganglia and globules. Heterogeneity of aquifer makes PCE migrate along preferential pathway. When PCE plume touches the zones of low permeability and high entry pressure, it will bypass these zones and migrate continuously, which leads to an increasing variability in PCE distribution. After PCE plume reaches the bottom of aquifer, PCE begin accumulate and form contaminant pool at the bottom. At t=100 days, a PCE pool is formed at the bottom of aquifer, moving toward the right boundary.

Figs. 6a-f show the simulated PCE saturation for individual realization of porous media for SPA during migration period. Under the effects of gravity force and natural hydraulic gradient, PCE is migrating and contaminant plume becomes larger and larger. Heterogeneity of aquifer significantly changes the migration paths and leads to irregular morphology of the PCE plume (Figs. 6a-c). However, due to the different

micro-arrangement of aquifer, the entry pressure distribution also is different which
leads to some differences. After the PCE injection, the simulated PCE saturation in
Figs. 6d-f indicates that further trapping and spreading of the PCE occurs during this
period. Compared with the simulation results of RTA in Fig. 5, the PCE plume slightly
seems similar in Fig. 6. Moreover, PCE infiltrates more quickly in porous media of
RTA in Fig. 5. After 70 days, PCE plume has touched the bottom for RTA (Fig. 5e),
while PCE plume based on SPA still keeps a significant distance from bottom (Fig. 6e).

To clean up the DNAPL, 4% surfactant solution is injected through two injection

wells at a constant rate of 80 $m^3$/day over 50 days. Afterwards, following water-flush
is applied during 150~300 day. The locations of injection and production wells are
presented in Fig. 3b. The production well is rightly installed at the location of the
PCE spill position and two injection wells are located 39 m to the left and right of the
production well. Figs. 5g-l show the PCE remediation results of individual realization
for RTA. During the early remediation period, the effect of cleaning up DNAPL is not
yet apparent (Figs. 5g-i). When the water flushing begins, the surfactant solution
circulates throughout the contaminated aquifer (Figs. 5j-l). At t=200 days, 237.01 $m^3$
PCE is removed from contaminated aquifer, occupying 79.00% of the total released
PCE (Fig. 5j). As time goes on, 268.30 $m^3$ PCE is removed from aquifer and
remediation efficiency reaches 89.43%.

The same surfactant remediation is also conducted for individual realization of

SPA. Compare with the remediation for RTA, the remediation effect is more apparent
for SPA (Figs. 6g-l). As the remediation processes, more DNAPL is removed and less
DNAPL is remained at the bottom of aquifer. At t=200 day, 267.68 $m^3$ PCE is
removed from contaminated aquifer and corresponding remediation efficiency rise to
89.23%. At t=300 day, 285.32 $m^3$ PCE is cleaned up and remediation efficiency
reaches 95.11%. From results of remediation, it is obvious that microstructure has
effect on remediation for macroscopic scale aquifer. Results suggest contaminated
aquifer of RTA is hard to clean up by surfactant remediation while SPA can improve
DNAPL remediation efficiency.

## 410    3.2.2 PCE migration and SGS realizations

PCE migration and remediation processes are simulated for 200 realizations of

porosity field for porous media of RTA and SPA. The variations of contaminant mass,
the ganglia-to-pool ratio (GTP) and moments of PCE plume versus time are presented
in Figs. 7a-h. During 0~30 day, the PCE in aquifer increases linearly at a constant rate
of 10 $m^3$/day (Fig. 7a), which corresponding to contaminant spill stage. Afterward, PCE
volume keeps constant during the second stage ranged 30~100 day, while PCE
volume in aquifer is reduced when surfactant is injected into aquifer. After surfactant
and water flushing the contaminated aquifer, most DNAPL is cleaned up. The residual
DNAPL mass remained in aquifer of 0.67 $m^3$-119.89 $m^3$ with a mean of 22.42 $m^3$ and
0.79 $m^3$-103.33 $m^3$ with a mean of 12.51 $m^3$ are achieve for 200 heterogeneous
realizations based on the RTA and SPA, respectively. The average remediation
efficiency of SPA is undoubtedly higher than RTA, indicating the aquifer of SPA is easier
to clean up. PCE plume architectures are quantified by measuring the ganglia-to-pool
ratio (GTP) in Fig. 7b. Over entire periods, curves of GTP value appear obvious
oscillations. Surfactant has the ability of promoting solubilization and mobilization of
DNAPL can reduce GTP value. As a result, when surfactant is injected at t=100 day, the
GTP value reduces quickly. When surfactant injection is end and water flushing begins,
the GTP value increases with steep flank slope. At last, GTP values reach 0.10-0.41 with
a mean of 0.21 and 0.15-0.42 with a mean of 0.28 for 200 heterogeneous realizations
based on the RTA and SPA, respectively.

Fig. 7c shows cumulative PCE removal from contaminated aquifer versus flushing

time for RTA and SPA. During the surfactant injection period ranged 100~150 day, the
DNAPL removal is not apparent, However, DNAPL is removed effectively and quickly
during water flushing period. Through long time remediation, the removal PCE from
contaminated aquifer reach 179.89 $m^3$-298.98 $m^3$ with a mean of 277.29 $m^3$ and 196.45
$m^3$-298.87 $m^3$ with a mean of 287.21 $m^3$ for 200 realizations based on RTA and SPA,
respectively. Average remediation efficiency of SPA (95.83%) is obvious higher than
average remediation efficiency of RTA (92.52%).

Fig. 7d shows the GTP value as a function of cumulative PCE removal for

contaminated aquifer. The GTP remains at a relatively low level before 30% of the
DNAPL is removed from aquifer. When 40% of the total 300 $m^3$ PCE are removed, GTP
values are increasing and corresponding curves appear a wave crest because the high
saturation zone of PCE plume are dissolved and turned into ganglia state. After the wave
crest, the GTP values decline quickly with steep flank slope due to PCE ganglia removal
through water flushing. At last, GTP values increase at the end of remediation process for
200 realizations, indicating most of PCE is removed and most of residual PCE turn to
ganglia state.
For the center of PCE plume in horizontal axis, associated variations versus time are
similar for 200 realizations based on RTA and SPA (Fig. 7e). Significantly, the PCE
plume vertical infiltration rate in aquifer of RTA is slightly faster than PCE infiltration in
aquifer of SPA for 200 realizations (Fig. 7f). Simultaneously, the second PCE plume
moments in horizontal direction of RTA are different from the second PCE plume
moments in horizontal direction of SPA (Fig. 7g). After PCE migration at natural
condition at t=100 day, the second PCE plume moments in horizontal direction are 10.61
$m^2$-40.50 $m^2$ with a mean of 21.51 $m^2$ and 10.99 $m^2$-36.38 $m^2$ with a mean of 20.75 $m^2$
for 200 realizations based on RTA and SPA, respectively. At t=300 day, the second PCE
plume moments in horizontal direction change to 0.81 $m^2$-34.88 $m^2$ with a mean of 5.79
$m^2$ and 1.03 $m^2$-24.57 $m^2$ with a mean of 4.64 $m^2$ for RTA and SPA, respectively. The
horizontal second moment of RTA is always larger than horizontal second moment of
SPA, indicating the PCE plume in aquifer of RTA is wider than PCE plume in aquifer of
SPA and RTA can cause larger range of groundwater contamination. Similarly, the second
moments in vertical direction for RTA are larger than the second moments in vertical
direction for SPA.
This study takes an important step toward exploring how micro-scale arrangements
control contaminant migration at small aquifer scale. Results are essential to the
macroscopic aquifer composed of porous media without large heterogeneity, such as
sandy aquifers containing rich groundwater resources. However, upscaling problem of
aquifer is widely existed in nature (Dagan et al., 2013; Pacheco, 2013; Pacheco et al.,
2015). Due to large heterogeneity of natural aquifers, research results may be very
different and can't be extrapolated to complex regional aquifer at large scale. On the
other hand, the finding in this study is absolutely applicable for natural aquifers with
similar heterogeneities. If the heterogeneity and anisotropy of natural aquifers are very
different, the effect of the micro-scale arrangements on the macroscopic contaminant
migration and remediation will be different. Even realistic conditions are complex, the
new findings achieved from this research also is very significant for understanding the
effect of micro-scale arrangement on contaminant behaviors at aquifer scale. The
upscaling problem of the results obtained at the simulation scale (100 x 25 x 25 m) is the
basis and the upscaling problem with more complex heterogeneity conditions is needed
to be further investigated. Various researches on upscaling problem are done from the
aspects of experiment and simulation (Wu et al., 2017a, 2017b, 2017c, 2017d). Based on
these research, the microstructure of porous media is developed and the contaminates
migration in porous media are explored using fractal methods in this study, implying the
experimental results are very significant for realistic problems at aquifer scale. Our next
procedure is applying these models in realistic aquifer with complex heterogeneity
conditions and modifying our models and method according to realistic conditions.

## 4. Conclusions

The micro-structure of aquifer has important effect on macroscopic scale
characteristics of contaminant migration and remediation. In this study, we focus on the
DNAPL migration and remediation in heterogeneous aquifers composed of granular
porous media with RTA and SPA. The microscale models of RTA and SPA are developed
to obtain the mathematical expressions of permeability and entry pressure using fractal
method. 200 realizations of porosity field are generated using SGS method and PCE is
released from underground storage tank into heterogeneous aquifer. To clean up
contamination caused by underground storage tank spill, surfactant remediation
technique is used to remove contaminants in aquifer. The entire process of DNAPL
migration and remediation is simulated by a multicomponent, multiphase model
simulator UTCHEM. Results suggest RTA not only cause larger range of groundwater
contamination than RTA, but also the contaminated aquifer of RTA is harder to clean up
compared with SPA. The second PCE plume moments in horizontal direction are 10.61
$m^2$-40.50 $m^2$ with a mean of 21.51 $m^2$ and 10.98 $m^2$-36.38 $m^2$ with a mean of 20.75 $m^2$
for 200 realizations based on RTA and SPA after long-term migration at t=100 day,
respectively. Furthermore, the second PCE plume moments in horizontal direction at
t=300 day are 0.807 $m^2$-34.88 $m^2$ with a mean of 5.79 $m^2$ and 1.025 $m^2$-24.57 $m^2$ with a
mean of 4.64 $m^2$ for RTA and SPA respectively after long-term remediation.
Simultaneously, the residual DNAPL mass remained in aquifer are 0.67 $m^3$-119.89 $m^3$
with a mean of 22.42 $m^3$ and 0.79 $m^3$-103.33 $m^3$ with a mean of 12.51 $m^3$ for RTA and
SPA respectively, indicating remediation efficiency of SPA (65.53%-99.74% with a mean
of 95.83%) mostly is higher than remediation efficiency of RTA (60.01%-99.78% with a
mean of 92.52%). This study reveals the microstructure of aquifer has important effect on
contaminant movement and associated remediation efficiency at macroscopic scale,

which is very essential and significant for dealing with the accidental event of underground storage tank spill and identifying subsurface contaminant source in the future.

## Acknowledgments

This research was financially supported by the National Key Research and Development Plan of China (2016YFC0402800), the National Natural Science Foundation of China (41772254 and 41372235), and the National Natural Science Foundation of China-Xianjiang project (U1503282). The authors are also profoundly grateful to Dr. Pacheco FLA and anonymous reviewer whose precious suggestions and constructive comments helped to improve the manuscript significantly.

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

**Table 1.** Parameters used in simulation

| Parameter | Value |
|---|---|
| Average value of porosity | 0.22 |
| Standard deviation of porosity | 0.06 |
| Longitudinal dispersivity | 1.0 m |
| Transverse dispersivity | 0.1 m |
| Hydraulic gradient | 0.005 m/m |
| Water density | 1.00 g/cm$^3$ |
| PCE density | 1.63 g/cm$^3$ |
| Surfactant density | 1.15 g/cm$^3$ |
| Water viscosity | 1.00 cp |
| PCE viscosity | 0.89 cp |
| PCE/ Water interfacial tension | 45 dyn/cm |
| PCE solubility in water | 240 mg/L |
| Residual water saturation | 0.24 |
| Residual PCE saturation | 0.17 |
| Endpoint of Water (BC model) | 0.486 |
| Endpoint of PCE (BC model) | 0.65 |
| Exponent of Water (BC model) | 2.85 |
| Exponent of PCE (BC model) | 2.7 |
| Exponent of capillary pressure | -0.52 |



 **Figure Captions**


**Figure 1.** Two different microscale arrangements of solid particles: (a) RTA; and (b)
SPA
**Figure 2.** Three kinds of Sierpinkski gasket [30]: (a) $L_a$=2; (b) $L_a$=3; and (c) $L_a$=5
**Figure 3.** (a) Two-dimensional view of contaminated domain; and (b) locations of
injection and extraction wells
**Figure 4.** (a) The individual porosity field generated by Sequential Gaussian Simulation
(SGS) method; (b) the frequency of individual porosity field; (c) the individual
permeability field of RTA obtained from individual porosity field; (d) the
frequency of individual permeability field for RTA; (e) the individual
permeability field of SPA obtained from individual porosity field; (f) the
frequency of individual permeability field for SPA; (g) The obtained individual
entry pressure field of RTA; (h) the frequency of individual entry pressure field
of RTA; (i) the obtained individual entry pressure field of SPA; and (j) the
frequency of individual entry pressure of SPA
**Figure 5.** Simulated PCE saturation for individual realization of RTA over the entire
migration and remediation periods ( 0~300 day)
**Figure 6.** Simulated PCE saturation for individual realization of SPA over the entire
migration and remediation periods ( 0~300 day)
**Figure 7.** (a) PCE volume in aquifer versus time, RTA represents RTA and SPA
represents SPA; (b) Changes in GTP as a function of time; (c) Cumulative
DNAPL removal as a function of time; (d) Variation of GTP value as a function
of cumulative DNAPL removal percent; (e) the change of the center of PCE
plume during the entire periods of migration and remediation; (f) the change of
the depth of PCE plume center during the entire periods; (g) variation of second
PCE plume moment in horizontal axis; and (h) variation of second PCE plume
moment in vertical axis


**Figure 1**

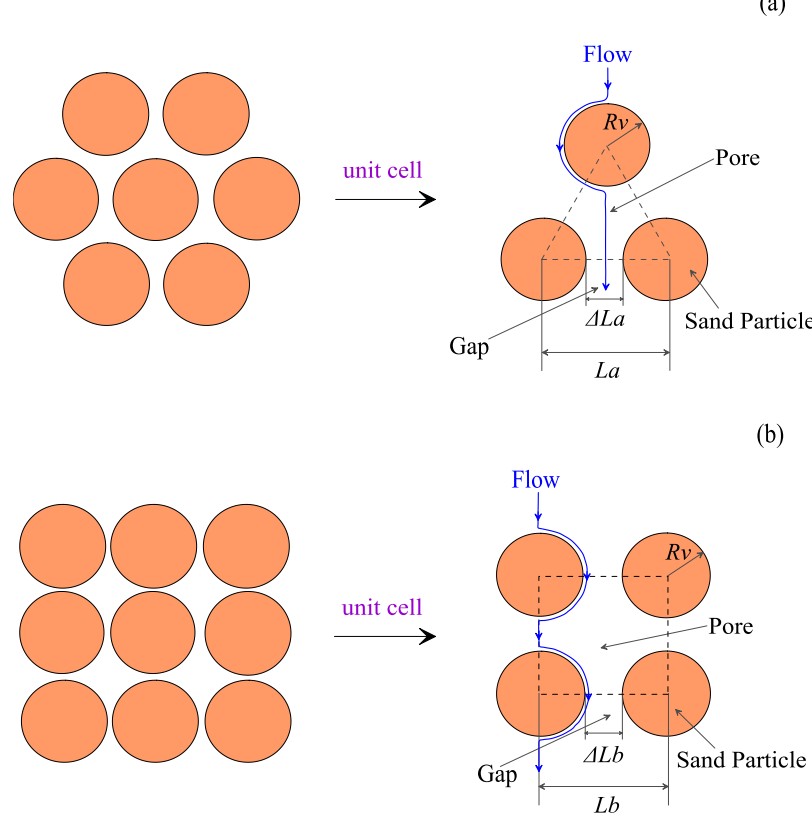



**Figure 2**

(a)  (b)  (c)

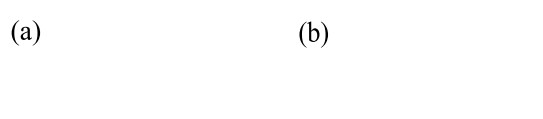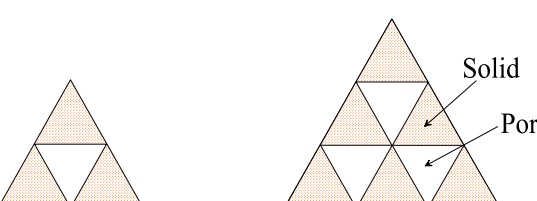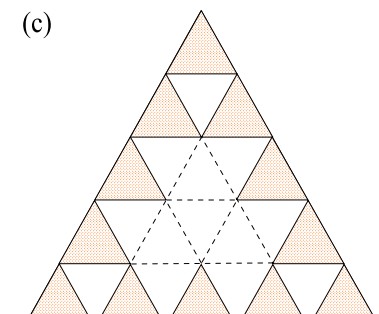




**Figure 3**

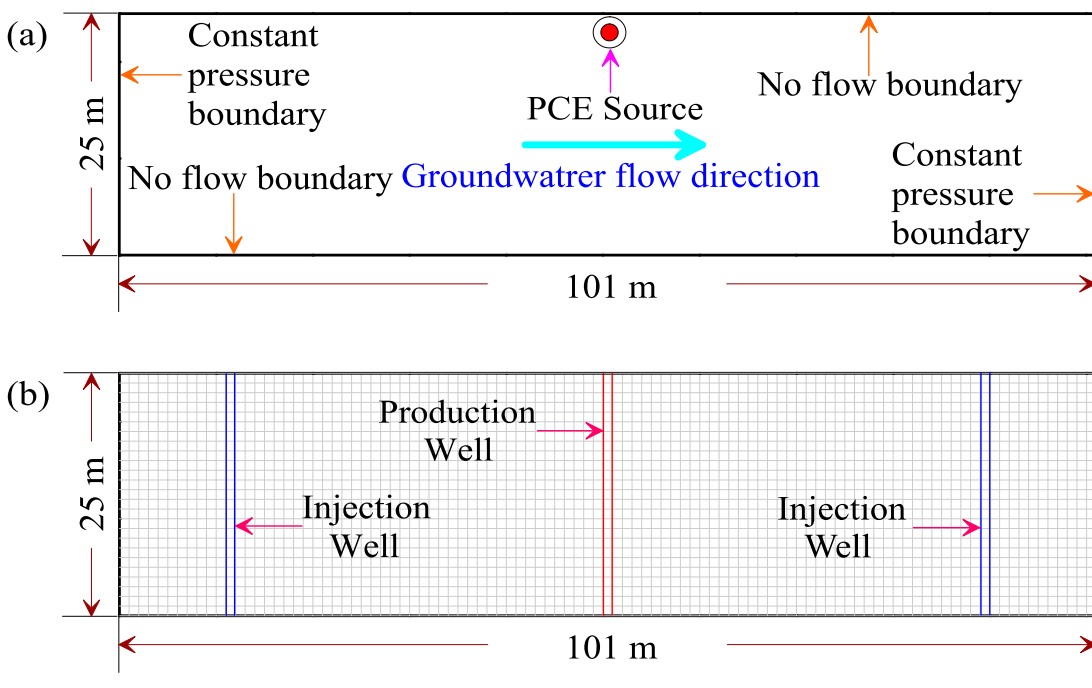



**Figure 4**

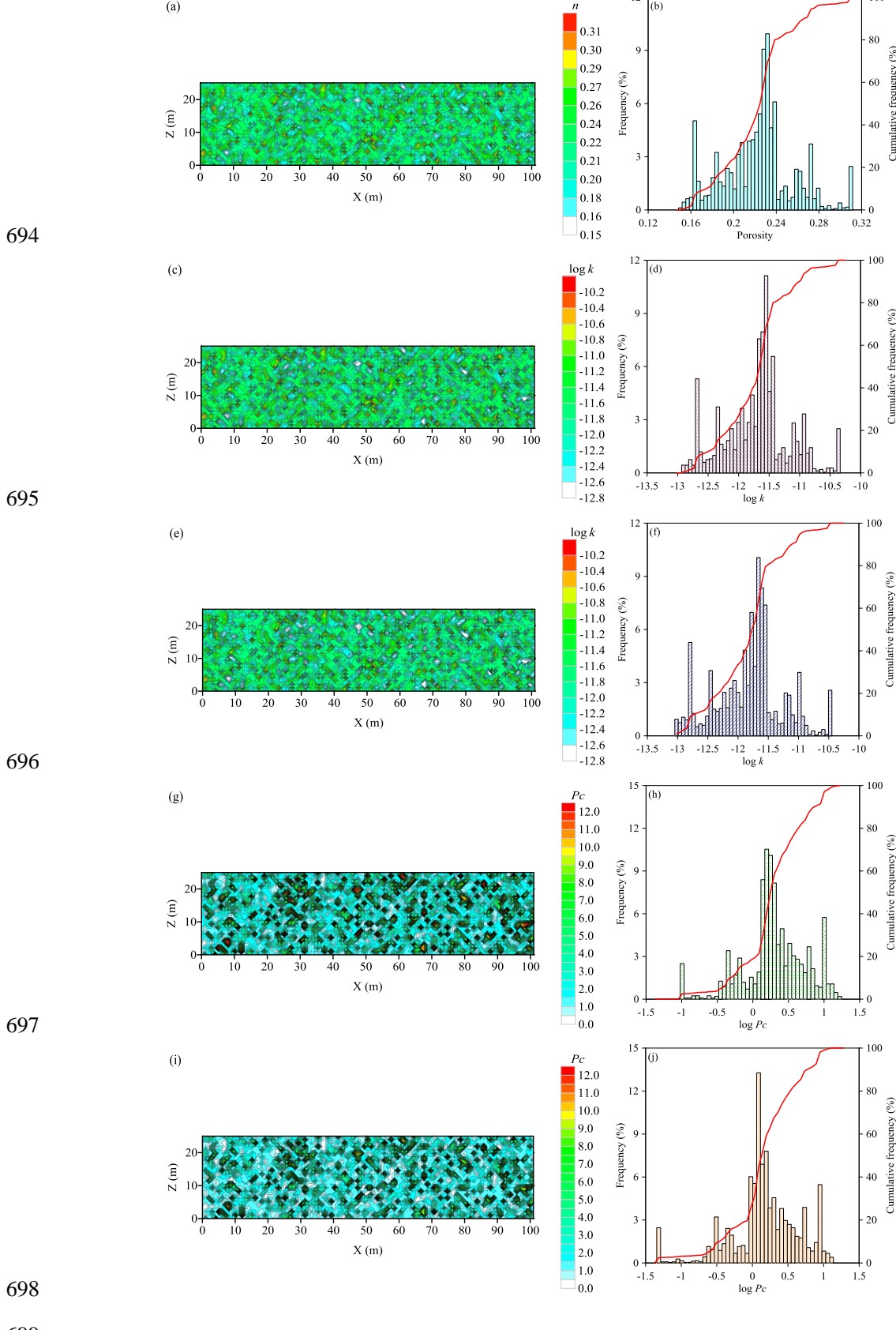







**Figure 5**

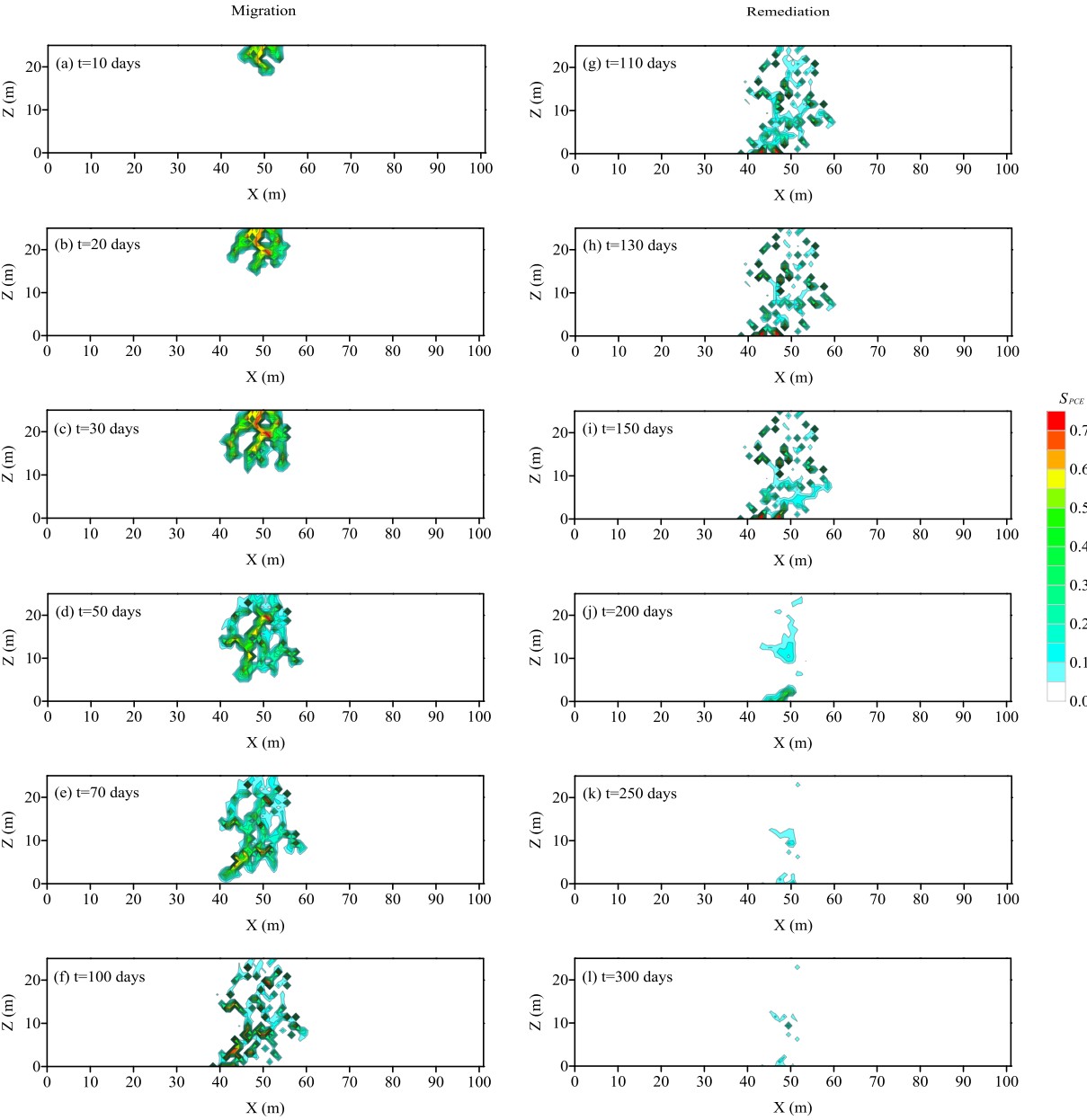

**Figure 6**

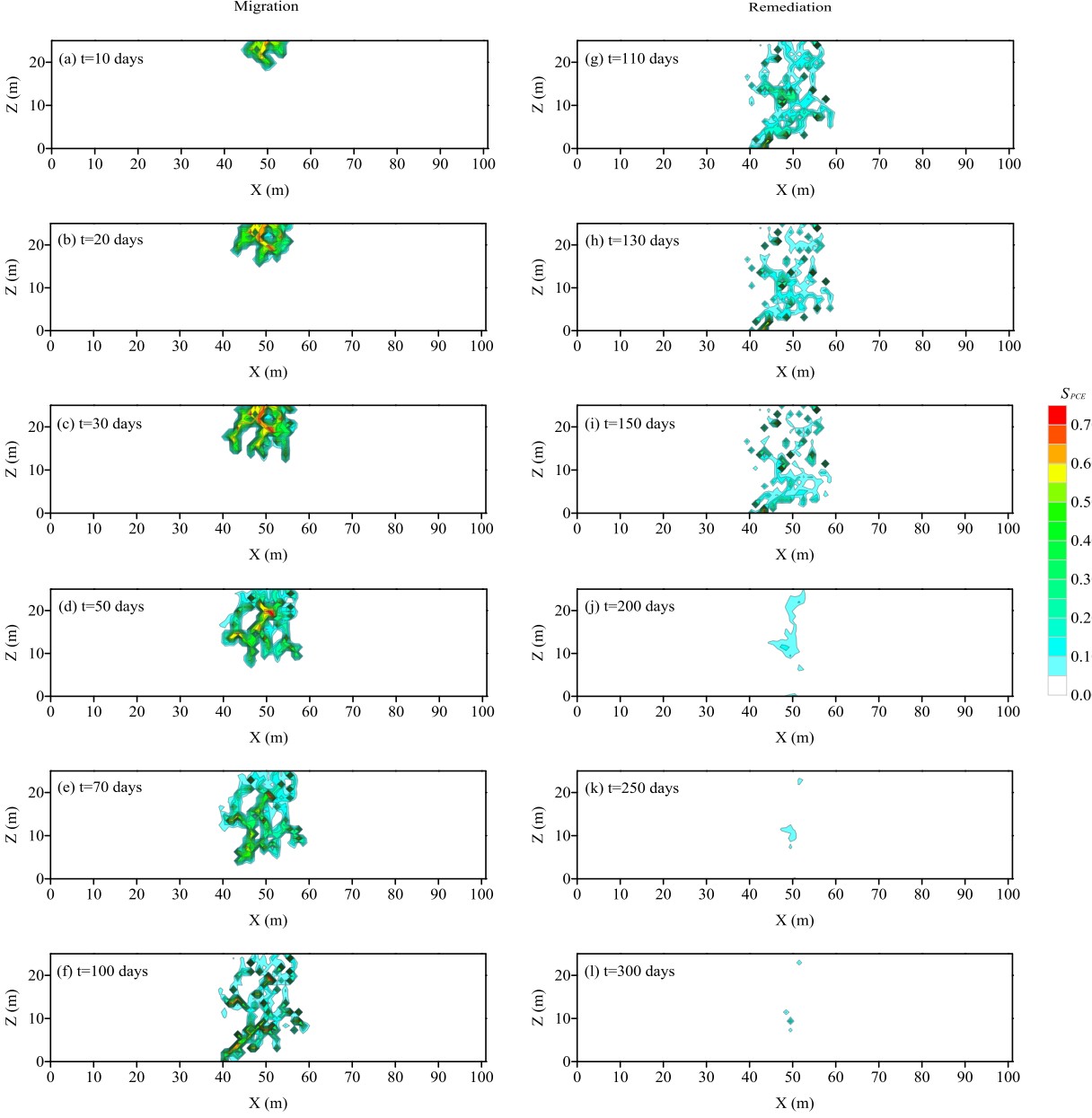



**Figure 7**

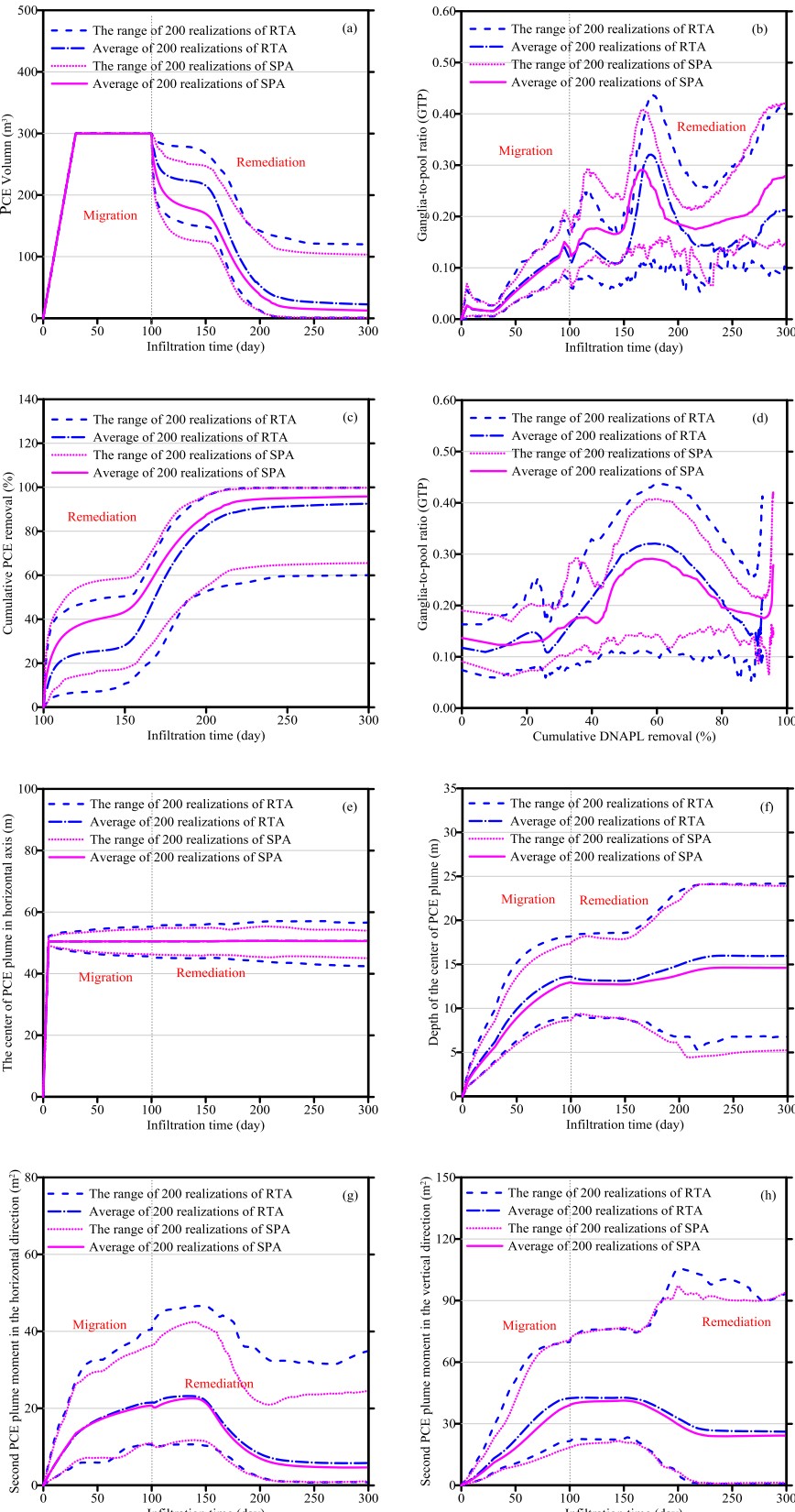