# Peer review of "Effects of Micro-Arrangement of Solid Particles on PCE"

_Hydrology and Earth System Sciences, 2017_

## Referee Comment (RC1) · F. Pacheco (Referee) · 21 Sep 2017

REVISION Special Issue Article: HESS-2017-493 Title: Effects of Micro-Arrangement of Solid Particles on PCE Migration and Its Remediation in Porous Media

Corresponding author: Jianfeng Wu Reviewer: Fernando A.L. Pacheco (fpacheco@utad.pt)

OUTLINE AND GENERAL APPRECIATION This study presents the results of a contaminant transport simulation based on the migration of DNAPL through an idealized porous media aquifer, which aimed to explore the control of porous media microstructure (micro arrangement of particles) over the migration and remediation of DNAPL at the macroscopic scale. The manuscript is well organized and written. Figures and tables are of good quality and are all necessary. The paper merits publication in the HESS, with a minor revision focused on issues of field validation and scale. All the conceptual and mathematical frameworks are correct, but to my view results cannot be extrapolated to real world conditions because no field validation is provided. Besides, heterogeneity of natural aquifers largely exceeds the simulation scale (100 x 25 x 25 m). So, although the effect of scale is discussed in the paper, to my view the results presented by the authors are valid solely for the heterogeneity conditions generated within the simulation, which means they are not directly transposed to larger field scales (see for example Pacheco, 2013 and Pacheco et al., 2015). In the discussion section of the revised manuscript, the authors should explain how they think their results can become representative at the aquifer scale, where heterogeneity and anisotropy are frequent, and how they couple with published results for natural systems. In fact, I already reviewed various manuscripts of this author and I think never saw this recommendation attended.

REFERENCES

Pacheco, F.A.L. (2013). Hydraulic diffusivity and macrodispersivity calculations imbedded in a geographic information system. Hydrological Sciences Journal, v. 58, no 4, p. 930-944.

Pacheco, F.A.L., Landim, P.M.B., & Szocs, T. (2015). Bridging hydraulic diffusivity from aquifer to particles size scale: A study on loessy sediments from SW Hungary. Hydrological Sciences Journal, v. 60(2), p. 269-284.

RECOMMENDATION Minor revision 21 September 2017 Fernando A.L. Pacheco
* * *

---

## Author Comment (AC1) · 29 Oct 2017

Comments accepted .   We appreciate Dr.  Pacheco's conscientious and positive recommendation.  We have made great efforts to implement the "minor revision" to improve our manuscript as suggested.  This research aims at characterizing the important effect of micro-structure on macroscopic scale characteristics of aquifer and inner contaminant migration and remediation. To achieve the aim of this research, the first step is to explore the issue of theory and simulation.  Fractal models of regular triangle arrangement (RTA) and square pitch arrangement (SPA) at microscale are developed to evaluate contaminant movement and associated contamination reme-

diation for a synthetic heterogeneous PCE contaminated site. Results suggest the micro arrangement of particles has obvious effects on macroscopic PCE migration and remediation. As for the upscaling problem of the results obtained at micro-scale, it is very important and interesting, and many previously published studies [e.g., Pacheco et al., 2013, 2015; Dagan et al., 2013; Bakshevskaia and Pozdniakov, 2016] are available and worth referencing. The upscaling problem of the results obtained at the simulation scale ($100 \times 25 \times 25$ m) is the basic research and the upscaling problem with more complex heterogeneities needs to be further investigated. We have made some progresses on upscaling problem from the aspects of experiment and simulation. On this basis, we have developed the microstructure of porous media and characterized the contaminant migration in porous media using fractal methods. So our next step is to apply these methods to real-world aquifers with highly heterogeneous conditions and to update our models according to general hydrogeological conditions. Considering Dr. Pacheco's concern, we have made additional and necessary explanation of upscaling issue in the section of Results and Discussion (lines 465-487) .

Please also note the supplement to this comment:
https://www.hydrol-earth-syst-sci-discuss.net/hess-2017-493/hess-2017-493-AC1-supplement.pdf

[Figure]

**Supplement:**

[revised manuscript text omitted]

(a)          (b)          (c)

[Figure]

[Figure]

Solid

Pore

**Figure 3**

(a)

[Figure]

(b)

[Figure]

**Figure 4**

[Figure]

**Figure 5**

[Figure]

**Figure 6**

[Figure]

**Figure 7**

[Figure]

---

## Referee Comment (RC2) · Anonymous Referee #2 · 16 Nov 2017

This paper introduced the effects of micro-structure on the contaminant migration and remediation. The results and discussion are clearly demonstrated. This paper met the quality requirement.

Language should be proofed by peers who were native speakers. Colloquial and informal words need to be revised. Some sentences were confused to understanding. Singular and plural problems and tense problems can be found in this paper. Please double check format requirements of the journal and whether case of every sentence is right

Uncertainty could be involved in real-world scenario; how did you treat the uncertainty

and error in the modeling?

Is that possible that the assumption of prorosity can be calibrated via BET analysis or other instrumental methods?

Following papers should be cited to improve this paper:

Shen, J., Huang, G., An, C., Zhao, S., & Rosendahl, S. (2017). Immobilization of tetra-bromobisphenol A by pinecone-derived biochars at solid-liquid interface: Synchrotron-assisted analysis and role of inorganic fertilizer ions. Chemical Engineering Journal, 321, 346-357. C. J. An, E. McBean, G. H. Huang, Y. Yao, P. Zhang, X. J. Chen and Y. P. Li. (2016). Multi-Soil-Layering Systems for Wastewater Treatment in Small and Remote Communities. Journal of Environmental Informatics, 27(2), 131-144. A. K. Mishra, B. Kumar and J. Dutta. (2016). Prediction of Hydraulic Conductivity of Soil Bentonite Mixture Using Hybrid-ANN Approach. Journal of Environmental Informatics, 27(2), 98-105.

————————————————————

---

## Author Response (AR1)

Response to Editor's and Reviewers' Comments on Manuscript hess-2017-493

**"Effects of Micro-Arrangement of Solid Particles on PCE Migration and**

**Its Remediation in Porous Media"** by Ming Wu, Jianfeng Wu, Jichun Wu, and Bill X. Hu

Note that the following text in Arial Narrow font denotes Editor's and Reviewers' comments and in

Times New Roman font denotes our response to the comments in the review. In our resubmission, the marked-up manuscript version is combined in this Author's response file. All changes to the original manuscript are indicated in the marked-up manuscript version. Also, in our marked-up manuscript version, marked in a green strikethrough font is the text that should be removed from the original manuscript and marked in a red font is the text that has been added to the revision. In addition, Line number(s) mentioned below is referred to as that line numbering in the marked revised manuscript.

## **Response to Dr. Pacheco FLA's Comments:**

OUTLINE AND GENERAL APPRECIATION

This study presents the results of a contaminant transport simulation based on the migration of DNAPL

through an idealized porous media aquifer, which aimed to explore the control of porous media microstruc-ture (micro arrangement of particles) over the migration and remediation of DNAPL at the macroscopic scale. The manuscript is well organized and written. Figures and tables are of good quality and are all necessary. The paper merits publication in the HESS, with a minor revision focused on issues of field validation and scale. All the conceptual and mathematical frameworks are correct, but to my view results cannot be extrapolated to real world conditions because no field validation is provided. Besides, heterogeneity of natural aquifers largely exceeds the simulation scale (100 x 25 x 25 m). So, although the effect of scale is discussed in the paper, to my view the results presented by the authors are valid solely for the heterogeneity conditions generated within the simulation, which means they are not directly transposed to larger field scales (see for example Pacheco, 2013 and Pacheco et al., 2015). In the discussion section of the revised manuscript, the authors should explain how they think their results can become representative at the aquifer scale, where heterogeneity and anisotropy are frequent, and how they couple with published results for natural systems. In fact, I already reviewed various manuscripts of this author and I think never saw this recommendation attended.

REFERENCES

Pacheco, F.A.L. (2013). Hydraulic diffusivity and macrodispersivity calculations imbedded in a geographic information system. Hydrological Sciences Journal, v. 58, no 4, p. 930-944.

Pacheco, F.A.L., Landim, P.M.B., & Szocs, T. (2015). Bridging hydraulic diffusivity from aquifer to particles size scale: A study on loessy sediments from SW Hungary. Hydrological Sciences Journal, v.

60(2), p. 269-284.

RECOMMENDATION Minor revision 21 September 2017 Fernando A.L. Pacheco

**[Response]** Comments partially accepted. We appreciate Dr. Pacheco's conscientious and positive recommendation. We have made great efforts to implement the "minor revision" to improve our manuscript as suggested. This research aims at characterizing the important effect of micro-structure on macroscopic scale characteristics of aquifer and inner contaminant migration and remediation. To achieve the aim of this research, the first step is to explore the issue of theory and simulation. Fractal models of regular triangle arrangement (RTA) and square pitch arrangement (SPA) at microscale are developed to evaluate contaminant movement and associated contamination remediation for a synthetic heterogeneous PCE

contaminated site. Results suggest the micro arrangement of particles has obvious effects on macroscopic PCE migration and remediation. As for the upscaling problem of the results obtained at micro-scale, it is very important and interesting, and many previously published studies [e.g., Pacheco et al., 2013, 2015; Dagan et al., 2013[†]; Bakshevskaia and Pozdniakov,

2016[††]] are available and worth referencing.

The upscaling problem of the results obtained at the simulation scale ($100 \times 25 \times 25$ m) is the basic research and the upscaling problem with more complex heterogeneities needs to be further investigated. We have made some progresses on upscaling problem from the aspects of experiment and simulation. On this basis, we have developed the microstructure of porous media and characterized the contaminant migration in porous media using fractal methods. So our next step is to apply these methods to real-world aquifers with highly heterogeneous conditions and to update our models according to general hydrogeological conditions.

Considering Dr. Pacheco's concern, we have made additional and necessary explanation of upscaling issue in the section of Results and Discussion (lines 554-574).

## **Response to Referee #2's Comments:**

OUTLINE AND GENERAL APPRECIATION

Anonymous Referee #2

This paper introduced the effects of micro-structure on the contaminant migration and remediation. The results and discussion are clearly demonstrated. This paper met the quality requirement. Language should be proofed by peers who were native speakers. Colloquial and informal words need to be revised.

Some sentences were confused to understanding. Singular and plural problems and tense problems can be found in this paper. Please double check format requirements of the journal and whether case of every sentence is right

**[Response]** We appreciate the Referee#2's positive comments and constructive suggestions.

Accordingly, we have made great efforts to improve the manuscript and fully incorporated the referee's suggestions into our revised manuscript. Moreover, we have thoroughly checked the manuscript to correct English grammar errors and expression.

Uncertainty could be involved in real-world scenario; how did you treat the uncertainty and error in the modeling?

**[Response]** Comments accepted. As the referee stated, uncertainty is widely existed in reality and should be considered in real-world scenario. As a result, in this study, Sequential Gaussian

Simulation (SGS) has been used for generating 200 random realizations of heterogeneous porosity field to deal with the uncertainty in the modeling.

Is that possible that the assumption of prorosity can be calibrated via BET analysis or other instrumental methods?

**[Response]** Yes. This study focuses on the effect of micro-arrangement of sand particles on macroscopic DNAPL migration and associated remediation for underground storage tank spill. Heterogeneous porosity distribution is generated using Sequential Gaussian Simulation (SGS) method. Afterwards, permeability and entry pressure are derived based on regular triangle arrangement (RTA) and square pitch microscale arrangement (SPA), respectively.

UTCHEM is then used to simulate the entire process of DNAPL migration and remediation in idealized heterogeneous contaminated site to reveal how the microstructure of porous media controls the contaminant migration and remediation at macroscopic scale. BET is a common technique used for determination of the surface area, porosity and other parameters of materials.

For realistic porous media, the porosity can be measured using BET analysis. However, this issue is beyond this study and will be explored in our further work.

Following papers should be cited to improve this paper:

Shen, J., Huang, G., An, C., Zhao, S., & Rosendahl, S. (2017). Immobilization of tetrabromobisphenol A

by pinecone-derived biochars at solid-liquid interface: Synchrotronassisted analysis and role of inorganic fertilizer ions. Chemical Engineering Journal, 321, 346-357.

C. J. An, E. McBean, G. H. Huang, Y. Yao, P. Zhang, X. J. Chen and Y. P. Li. (2016). Multi-Soil-Layering

Systems for Wastewater Treatment in Small and Remote Communities. Journal of Environmental

Informatics, 27(2), 131-144.

A. K. Mishra, B. Kumar and J. Dutta. (2016). Prediction of Hydraulic Conductivity of Soil Bentonite

Mixture Using Hybrid-ANN Approach. Journal of Environmental Informatics, 27(2), 98-105.

Interactive comment on Hydrol. Earth Syst. Sci. Discuss., https://doi.org/10.5194/hess-2017-

493, 2017.

**[Response]** Comments accepted and the references have been cited in the revised manuscript.

We are grateful to the referee's whose constructive suggestions have led to significant improvement of the manuscript.

[revised manuscript text omitted]

(a)                  (b)                          (c)

[Figure]

[Figure]

[Figure]

**Figure 3**

(a)

[Figure]

(b)

[Figure]

**Figure 4**

[Figure]

**Figure 5**

[Figure]

**Figure 6**

[Figure]

**Figure 7**

[Figure]